# Population structure, biogeography and transmissibility of *Mycobacterium tuberculosis*

Luca Freschi[1✉], Roger Vargas Jr.[1,2], Ashaque Husain[3], S. M. Mostofa Kamal[4], Alena Skrahina[5], Sabira Tahseen[6], Nazir Ismail[7,8], Anna Barbova[9], Stefan Niemann[10], Daniela Maria Cirillo[11], Anna S. Dean[12], Matteo Zignol[12] & Maha Reda Farhat[1,13✉]

*Mycobacterium tuberculosis* is a clonal pathogen proposed to have co-evolved with its human host for millennia, yet our understanding of its genomic diversity and biogeography remains incomplete. Here we use a combination of phylogenetics and dimensionality reduction to reevaluate the population structure of *M. tuberculosis*, providing an in-depth analysis of the ancient Indo-Oceanic Lineage 1 and the modern Central Asian Lineage 3, and expanding our understanding of Lineages 2 and 4. We assess sub-lineages using genomic sequences from 4939 pan-susceptible strains, and find 30 new genetically distinct clades that we validate in a dataset of 4645 independent isolates. We find a consistent geographically restricted or unrestricted pattern for 20 groups, including three groups of Lineage 1. The distribution of terminal branch lengths across the *M. tuberculosis* phylogeny supports the hypothesis of a higher transmissibility of Lineages 2 and 4, in comparison with Lineages 3 and 1, on a global scale. We define an expanded barcode of 95 single nucleotide substitutions that allows rapid identification of 69 *M. tuberculosis* sub-lineages and 26 additional internal groups. Our results paint a higher resolution picture of the *M. tuberculosis* phylogeny and biogeography.

[1] Department of Biomedical Informatics, Harvard Medical School, Boston, MA, USA. [2] Department of Systems Biology, Harvard Medical School, Boston, MA, USA. [3] Directorate General of Health Services, Ministry of Health and Family Welfare, Dhaka, Bangladesh. [4] Department of Pathology and Microbiology, National Institute of Diseases of the Chest and Hospital, Dhaka, Bangladesh. [5] Republican Scientific and Practical Centre for Pulmonology and Tuberculosis, Minsk, Belarus. [6] National Reference Laboratory, National Tuberculosis Control Programme, Islamabad, Pakistan. [7] National Institute for Communicable Diseases, Sandringham, South Africa. [8] Department of Medical Microbiology, University of Pretoria, Pretoria, South Africa. [9] Central Reference Laboratory on Tuberculosis Microbiological Diagnostics, Ministry of Health, Kiev, Ukraine. [10] Molecular and Experimental Mycobacteriology, Borstel Research Centre, Borstel, Germany. [11] Emerging Bacterial Pathogens Unit, IRCCS San Raffaele Scientific Institute, Milan, Italy. [12] Global Tuberculosis Programme, World Health Organization, Geneva, Switzerland. [13] Pulmonary and Critical Care Medicine, Massachusetts General Hospital, Boston, MA, USA. ✉email: l.freschi@gmail.com; maha_farhat@hms.harvard.edu

Tuberculosis (TB) is among the ten top causes of death worldwide. In 2018 more than ten million people fell ill and 1.5 million died from TB (https://www.who.int/tb/global-report-2019). During the last two decades, significant efforts have been made to understand strain-level genetic diversity in the TB bacillus *Mycobacterium tuberculosis* (*Mtb*) and its geographic distribution. A robust classification of *Mtb* strains into evolutionarily meaningful sub-lineages is important both for taxonomic purposes and because sub-lineages can differ in virulence or antibiotic resistance[1]. *Mtb* was first classified into three principal genetic groups in 1997, based on two neutral single nucleotide substitutions (SNSs) in the antibiotic resistance genes *katG* (codon 463) and *gyrA* (codon 95)[2]. Since then several studies have attempted higher resolution classification, using large genomic deletions[3], spoligotyping[4] and SNSs[5–11]. There are currently nine recognized *Mtb* lineages (L1–9), with two lineages (L2, L4) that have been well represented in taxonomic and phylogeographic evaluations, while two others (L1 and L3) were the subject of more recent works[6–9,12–17]. Within the major lineages, 53 sub-lineages have been described and are definable with a SNS "barcode"[6]: 7 for L1, 6 for L2, 4 for L3, and 36 for L4. These 53 sub-lineages best characterize diversity within L2 and L4. The population structure of L1 and L3 is less understood as these lineages are most prevalent in countries where pathogen sequencing had been less widely applied. Recently, studies fueled by increasing sequencing capacity in high-burden TB settings have begun to evaluate the evolutionary history of L1 and L3 including the role of migration and dispersal in driving their prevalence in different parts of the world[15–18].

Host-pathogen co-evolution and more recent host-related selective pressures have been postulated to drive genetic diversity in *Mtb*[19,20]. Of high relevance to public health, this genetic diversity is thought to underlie the observed differences in *Mtb* sub-lineage transmissibility[21,22] or host specificity[8]. For example, of the *Mtb* sub-lineages prevalent in Ho Chi Minh City, Vietnam, one modern L2 sub-lineage (2.2.1) was found to be more transmissible relative to other L2 sub-lineages, L4 or L1[21]. Another study reported reduced transmissibility for L3 compared to the other Mtb major lineages in Montreal, Canada[23]. Finally, the phylogeography of L5 and L6 (also known as *M. africanum*), L7, L8 and even some L4 sub-lineages (e.g., 4.6.1/Uganda, 4.6.2/Cameroon, 4.1.3/Ghana, and 4.5) has demonstrated that these groups are more geographically restricted than L2 and other L4 sub-lineages. These geographically restricted groups are associated with lower variation in *Mtb* T-cell epitopes and support the notion that *Mtb* sub-lineages have a spectrum of pathogenic strategies and maybe niche specialists infecting preferentially humans of a specific population or ancestry[8]. However, more evidence is needed to support this hypothesis given the relatively few genetic differences that define *Mtb* sub-lineages or human ancestry.

Here, we sought to expand our understanding of the *Mtb* population structure by leveraging 9584 genomes sampled from 49 countries, including 738 L1 and 1104 L3 isolates. We identify and validate 22 novel sub-lineages and 8 additional internal groups (i.e., genetically divergent groups found in sub-lineages that cannot be further partitioned in a hierarchical fashion according to our criteria), including 7 in L1 and 4 in L3, and expand the SNS typing barcode to 95 sites. We find that L3 and L2 have similar phylogenetic structure, yet L2 along with L4 manifest a phylogenetic signal of increased transmissibility compared with L1 or L3. These findings, along with our observations of novel geographically restricted sub-lineages, expand the evidence supporting the *Mtb* human co-evolution hypothesis.

## Results

**Detailed population structure of L1–4 and a hierarchical sub-lineage naming system**. We assembled a high-quality data set of whole genomes, antibiotic resistance phenotypes, and geographic sites of isolation for 9584 clinical *Mtb* samples ("Methods" section and Supplementary Data 1). Of the total, 4939 (52%) were pan-susceptible, i.e., susceptible to at least isoniazid and rifampicin (and all other antibiotics when additional phenotypic data were available), and 4645 (48%) were resistant to one or more antibiotics (Supplementary Fig. 1a). Using the 62 SNS lineage barcode[6], 738 isolates were classified as L1 (8%), 2193 as L2 (22%), 1104 as L3 (12%) and 5549 as L4 (58%, Supplementary Fig. 1b). Among the 4939 pan-susceptible isolates, we identified high-quality genome-wide SNSs (83,735 for L1, 56,736 for L2, 76,817 for L3, and 185,622 for L4) that we used in building maximum-likelihood phylogenies for each major lineage (L1–4, "Methods" section). We computed an index of genetic divergence ($F_{ST}$) between groups defined by each bifurcation in each phylogeny. Sub-lineages were defined as monophyletic groups that had high $F_{ST}$ (>0.33) and were also clearly separated from other groups in principal component analysis (PCA, see "Methods" section). We also defined internal groups to sub-lineages (see "Methods" section): an internal group is a monophyletic group genetically divergent (by $F_{ST}$ and PCA) from its neighboring groups, but has one or more ancestral branches that show a low degree of divergence or low support (bootstrap values). Internal groups do not represent true sub-lineages in a hierarchical fashion, but defining them allows us to further characterize the *Mtb* population structure. We provide code to automate all the steps described above. Our approach is scalable and can be used on other organisms (see "Methods" section).

To better classify *Mtb* isolates in the context of the global *Mtb* population structure, we developed a hierarchical sub-lineage naming scheme (Supplementary Data 2) where each subdivision in the classification corresponds to a split in the phylogenetic tree of each major *Mtb* lineage. Starting with the global *Mtb* lineage numbers (e.g., L1), we recursively introduced a subdivision (e.g., from 1.2 to 1.2.1 and 1.2.2) at each bifurcation of the phylogenetic tree whenever both subclades sufficiently diverged. Formally, we defined these splits using bootstrap criteria, and independent validations by $F_{ST}$ and PCA (see "Methods" section). Internal groups were denoted with the letter "i" (e.g., 4.1.i1). This proposed system overcomes two major shortcomings of the existing schemas: same-level sub-lineages are never overlapping (unlike the system of Stucki et al.[8] sub-lineage 4.10 includes sub-lineages 4.7–4.9), and the names reflect both phylogenetic relationships and genetic similarity (unlike *semantic* naming such as the "Asia ancestral" lineage in the system of Shitikov et al.[7]). Further, this naming system can be standardized to automate the process of lineage definition. These advantages come at the price of long sub-lineage names in the case of complex phylogenies (e.g., for L4, sub-lineage 4.10 gets the lineage designation 4.2.1.1.1.1.1.1). For compatibility with naming conventions already in use and to keep names as short as possible, we designed a second, shorthand, naming system that expands the Coll et al. lineage schema by adding new subdivisions and differentiating between sub-lineages and internal groups. For instance, sub-lineage 4.3.1 is designated as 4.3.i1, informing the user that this is an internal group of sub-lineage 4.3. To simplify the use of the hierarchical naming schema and the updated shorthand schema, we provide a table that compares them side by side along with naming systems currently in use (Supplementary Data 2).

Using the sub-lineage definition rules and the sub-lineage naming scheme described above, we characterized six previously

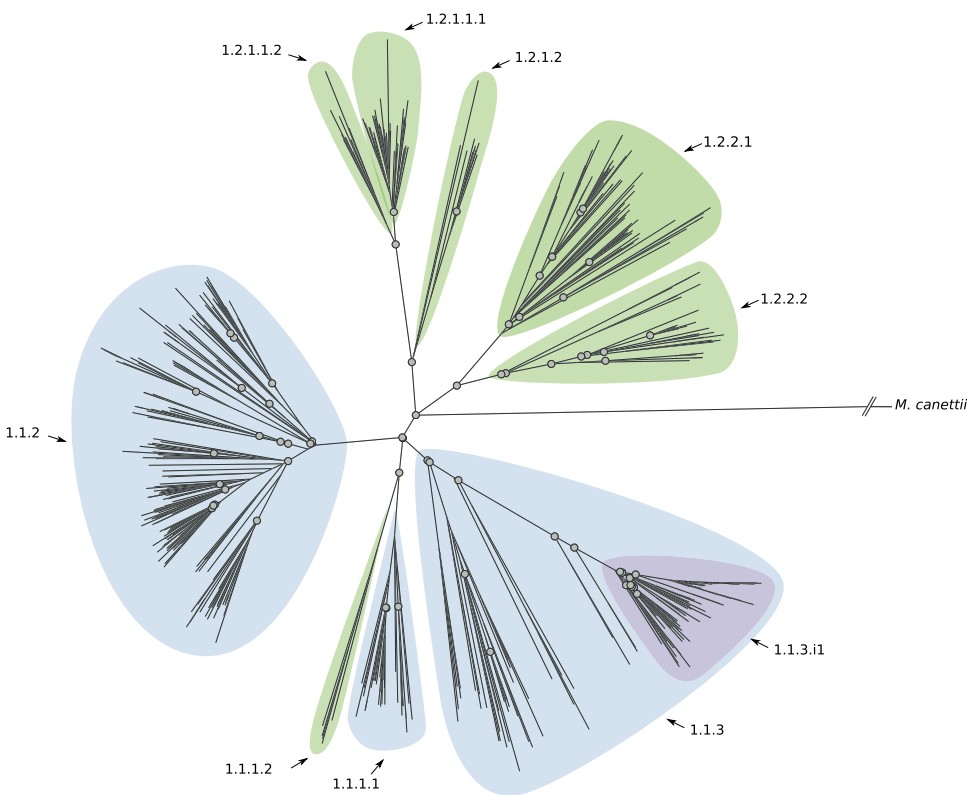

**Fig. 1 Phylogenetic tree reconstruction of lineage 1 (binary tree).** Gray circles define splits where the $F_{ST}$ (fixation index) calculated using the descendants of the two children nodes is greater than 0.33. The sub-lineages are defined by colored areas (blue: sub-lineages already described in the literature; green: sub-lineages described here; purple: internal sub-lineages). Source data are provided as a Source Data file.

undescribed sub-lineages of L1 (Fig. 1 and Supplementary Fig. 2); five of which expand the current description of 1.2. We also detected an internal group of 91 isolates (1.1.3.i1) characterized by a long defining branch in the phylogeny (corresponding to 82 SNSs), a high $F_{ST}$ (0.48), and geographically restricted to Malawi (85/91, 93% isolates, Fig. 1 and Supplementary Fig. 3). We estimated the date of the emergence of the MRCA of such a group (see "Methods" section) and we found it to be between 1497 and 1754. We found four previously undescribed sub-lineages of L3 (Fig. 2 and Supplementary Fig. 4), revising L3 into four main groups, whereas previously only two partitions of one sub-lineage were characterized (3.1). We found that the latter two partitions are in fact internal groups of the largest sub-lineage (3.1.1) in our revised classification.

L2 is divided into two groups: proto-Beijing and Beijing with the latter in turn partitioned into two groups: ancient- and modern-Beijing[7]. Each one of these groups is characterized by further subdivisions (three for the ancient-Beijing group and seven for the modern-Beijing group; see Supplementary Fig. 4). We found a new sub-lineage (2.2.1.2, Fig. 3, and Supplementary Fig. 5) within the previously characterized ancient-Beijing group. However, genetic diversity within the modern-Beijing group (2.2.1.1.1) was lower than in the other L2 sub-lineages and the tree topology and $F_{ST}$ calculations did not support further hierarchical subdivisions. Although we did find three internal groups of modern-Beijing: two undescribed and one that corresponds to the Central Asia group[7]. For L4, our results support a complex population structure with 21 sub-lineages and 15 internal groups. In particular, we found 11 previously undescribed sub-lineages and 5 internal groups that expand our understanding of previously characterized sub-lineages (e.g., 4.2.2; 4.2 in the Coll et al. classification) or that were not characterized since these isolates were simply classified as

L4 (e.g., 4.11, Fig. 4, and Supplementary Fig. 6) using the other barcodes.

**A new barcode to define L1–4 *Mtb* sub-lineages and a software package to type *Mtb* strains from WGS data.** We defined a SNS barcode for distinguishing the obtained sub-lineages (Supplementary Data 3). We characterized new synonymous SNSs found in 100% of isolates from a given sub-lineage, but not in other isolates from the same major lineage, compiling 95 SNSs into an expanded barcode (Supplementary Data 3). We validated the barcode by using it to call sub-lineages in the hold-out set of 4645 resistant isolates and comparing the resulting sub-lineage designations with maximum-likelihood phylogenies inferred from the full SNS data (Supplementary Figs. 7–10). A sub-lineage was validated if it was found in the hold-out data and formed a monophyletic group in the phylogeny. Considering the "recent" sub-lineages, i.e., the most detailed level of classification in our system, we were able to validate eight out of nine L1 sub-lineages including five out of six of the new sub-lineages described here, with the exception of 1.1.1.2. We validated all four new L3 sub-lineages, all five L2 sub-lineages including the one previously undescribed, and 16 of the 21 L4 sub-lineages including two described here. The sub-lineages we could not confirm were not represented by any isolate in the validation phylogenies. We did not observe any paraphyletic sub-lineages in the revised classification system.

We developed fast-lineage-caller, a software tool that classifies *Mtb* genomes using the SNS barcode proposed above. For a given genome, it returns the corresponding sub-lineage as output using our hierarchical naming system in addition to four other existing numerical/semantic naming systems, when applicable (see "Methods" section). The tool also informs the user on how many SNSs support a given lineage call and allows for filtering of

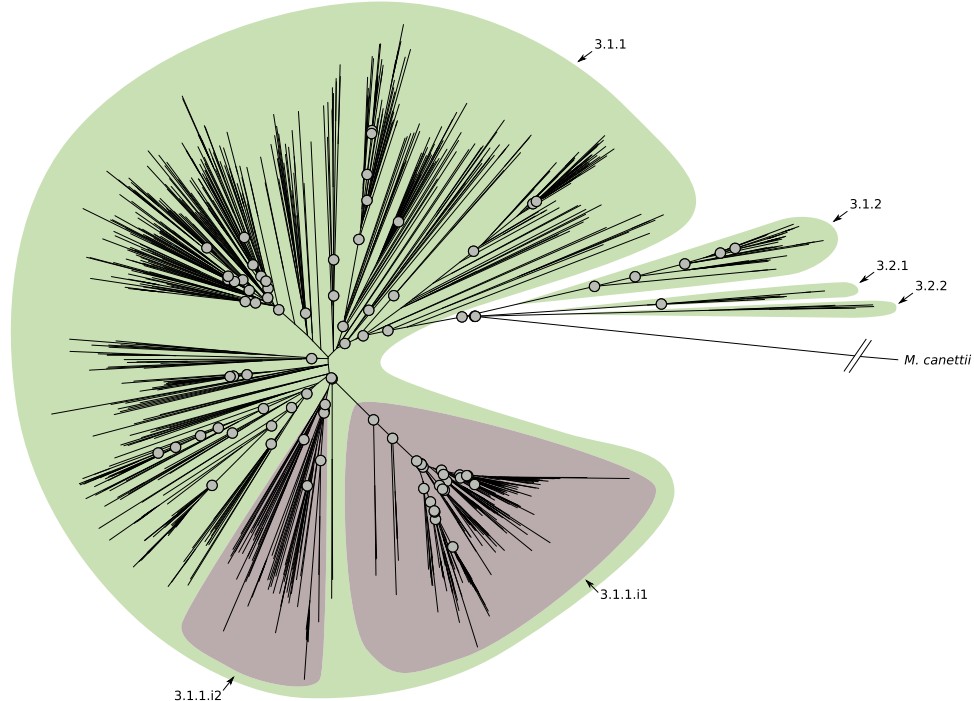

**Fig. 2 Phylogenetic tree reconstruction of lineage 3 (binary tree).** Gray circles define splits where the $F_{ST}$ (fixation index) calculated using the descendants of the two children nodes is greater than 0.33. The sub-lineages are defined by colored areas (green: sub-lineages described here; purple: internal sub-lineages). Source data are provided as a Source Data file.

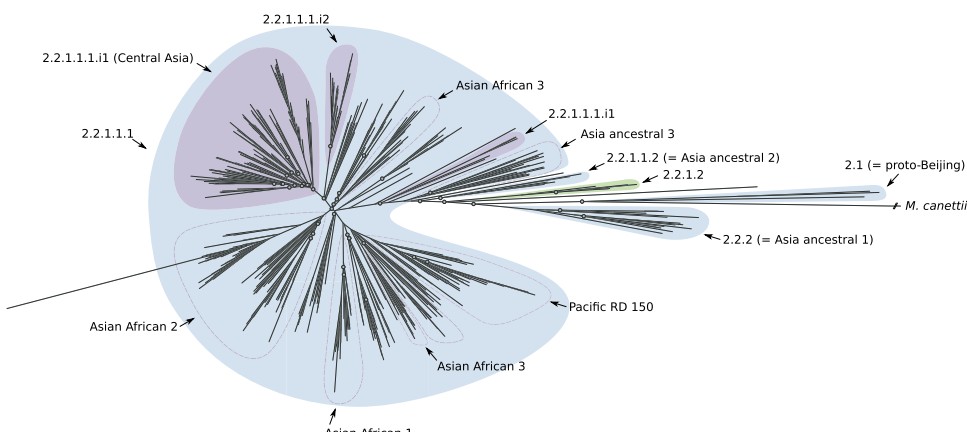

**Fig. 3 Phylogenetic tree reconstruction of lineage 2 (binary tree).** Gray circles define splits where the $F_{ST}$ (fixation index) calculated using the descendants of the two children nodes is greater than 0.33. The sub-lineages are defined by colored areas (blue: sub-lineages already described in the literature; green: sub-lineages described here; purple: internal sub-lineages). Source data are provided as a Source Data file.

low-quality variants. The tool is generalizable and can manage additional barcodes defined by the user to type the core genome of potentially any bacterial species.

**Geographic distribution of the *Mtb* sub-lineages.** Next, we examined whether certain sub-lineages were geographically restricted, which would support the *Mtb*-human co-evolution hypothesis, or whether they constituted prevalent circulating sub-lineages in several different countries (i.e., geographically unrestricted)[8]. We used our SNS barcode to determine the sub-lineages of 17,432 isolates (see "Methods" section) sampled from 74 countries (Supplementary Fig. 11 and Supplementary Data 4, 5). We computed the Simpson diversity index (Sdi) as a measure of geographic diversity that controls for variable sub-lineage frequency (see "Methods" section) for each well-represented sub-lineage or internal group ($n > 20$). We hypothesized that geographically

unrestricted lineages would have a higher Sdi. We found Sdi to correlate highly ($\rho = 0.68$; $p$-value $= 5.7 \times 10^{-7}$) with the number of continents from which a given sub-lineage was isolated (Supplementary Fig. 12). The Sdi ranged between a minimum of 0.05 and a maximum of 0.72, with a median value of 0.46 (Fig. 5). The known geographically restricted sub-lineages[8] had an Sdi between 0.28 and 0.5 (Fig. 5 and Supplementary Table 1), while the known geographically unrestricted sub-lineages[8,9] had an Sdi between 0.55 and 0.61 (Fig. 5 and Supplementary Table 2). We found 11 sub-lineages/internal groups with Sdi <0.28 (Supplementary Table 3), and 11 sub-lineages/internal groups with Sdi >0.61 (Supplementary Table 4), i.e., more extreme than previously reported geographically restricted or unrestricted sub-lineages, respectively.

While the currently known geographically restricted sub-lineages are all in L4, we found evidence of geographic restriction for two sub-lineages/internal groups of L1. The first, the L1

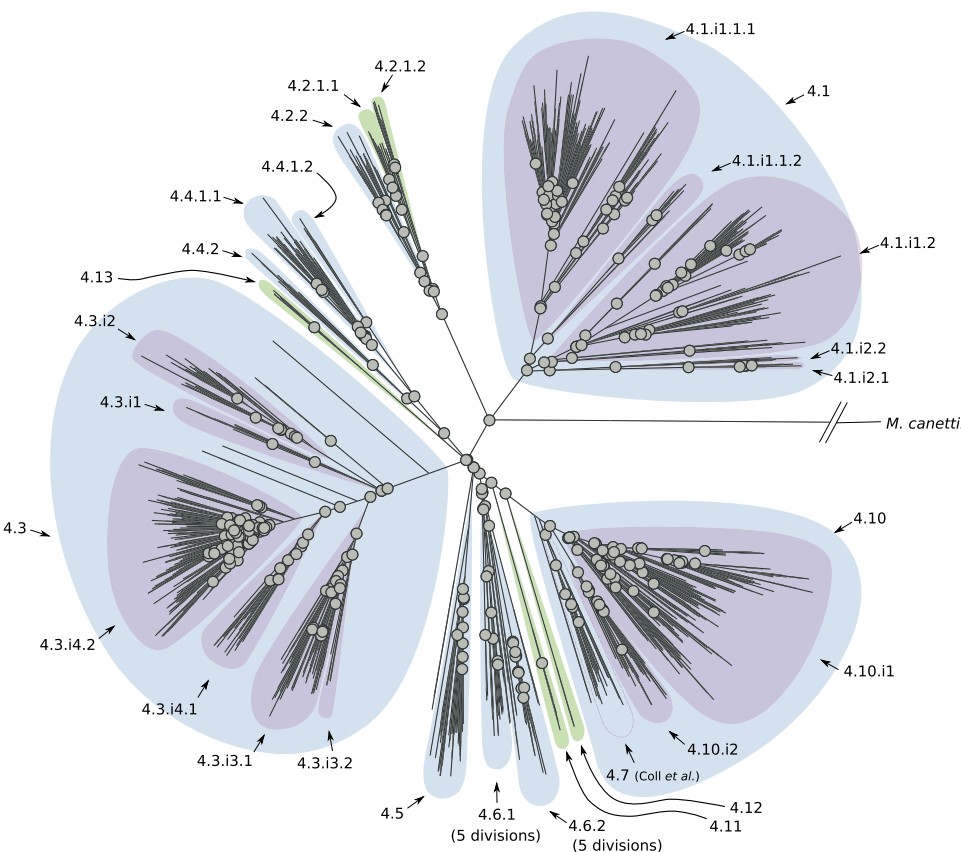

**Fig. 4 Phylogenetic tree reconstruction of lineage 4 (binary tree).** Gray circles define splits where the $F_{ST}$ (fixation index) calculated using the descendants of the two children nodes is greater than 0.33. The sub-lineages are defined by colored areas (blue: sub-lineages already described in the literature; green: sub-lineages described here; purple: internal sub-lineages). Source data are provided as a Source Data file.

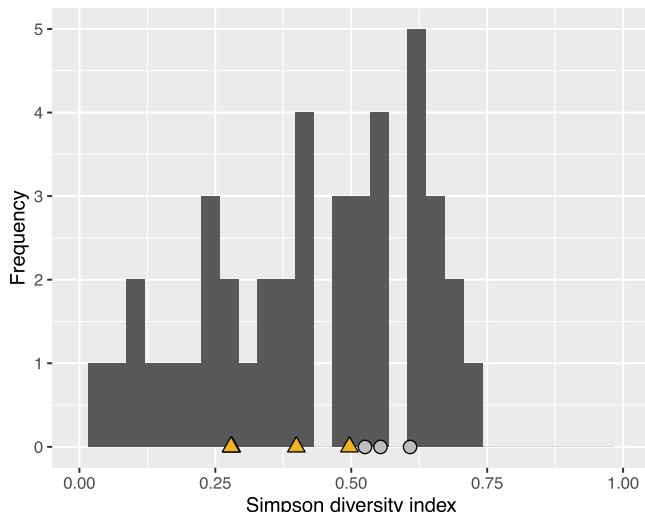

**Fig. 5 Histogram of the Simpson diversity index calculated for sub-lineages of lineages 1–4.** A data set of 17,432 isolates from 74 countries was used to perform this analysis. Yellow triangles designate the Simpson diversity index values of sub-lineages designated as geographically restricted by Stucki et al. Light gray circles designate the Simpson diversity index values of sub-lineages designated as geographically unrestricted by Stucki et al. Source data are provided as a Source Data file.

internal group 1.1.3.i1, showed a very low Sdi (0.06) and was only found at high frequency among the circulating L1 isolates in Malawi (Fig. 6). This finding is also in agreement with the L1 phylogeny (Fig. 1) that shows a relatively long (82 SNS)

branch defining this group. The second geographically restricted L1 sub-lineage is 1.1.1.1 (Sdi = 0.12) that was only found at high frequency among circulating L1 isolates in South-East Asia (Vietnam and Thailand, Fig. 7). To exclude the possibility that these two groups appeared geographically restricted as a result of oversampling transmission outbreaks, we calculated the distribution of the pairwise SNS distance for each of these two sub-lineages. We measured a median SNS distance of 204 and 401, respectively, refuting this kind of sampling error for these groups (typical pairwise SNS distance in outbreaks <15–20 SNS[24]) (Supplementary Fig. 13).

The geographically unrestricted sub-lineages spanned several recognized generalist sub-lineages or internal groups (e.g., 4.1.i1.1.1.1, 4.3, 4.10, and 2.2.1.1.1, which correspond to 4.1.2/ Haarlem, 4.3/LAM, 4.10/PGG3, modern-Beijing in the Stucki et al.[8] or the Shitikov et al.[7] classification). In addition, we found a candidate geographically unrestricted sub-lineage of L1 (1.1.2). L1.1.2 spanned 253 isolates from 7 countries and 4 continents and its Sdi was 0.61 (Fig. 8). Overall we found a low, but significant correlation between Sdi and the number of isolates sampled from each sub-lineage (ρ = 0.34; p-value = 0.03, Supplementary Fig. 14) indicating that the most prevalent sub-lineages were more likely to be geographically unrestricted.

To validate the geographic distribution of sub-lineages, we curated a data set of 3791 *Mtb* isolates sampled to reliably represent the entire population of TB patients in five countries (Azerbaijan, Bangladesh, Pakistan, South Africa, and Ukraine, Supplementary Data 6, Zignol et al. data set). In this data set, we were able to identify 9 of the 30 new groups/sub-lineages described above. L1 and 1.1.2 isolates were found in three out of five countries (Azerbaijan, Pakistan, and Ukraine) spanning two

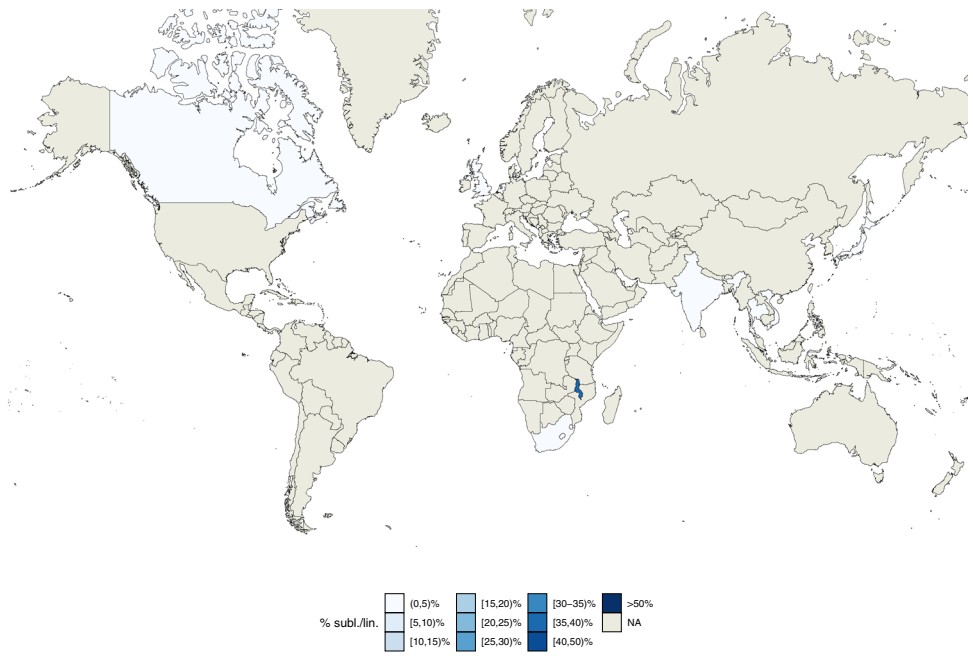

**Fig. 6 Geographic distribution of internal sub-lineage 1.1.3.i1.** Colors represent the percentage of 1.1.3.i1 strains isolated in a given country with respect to all lineage 1 strains isolated in such country. Source data are provided as a Source Data file.

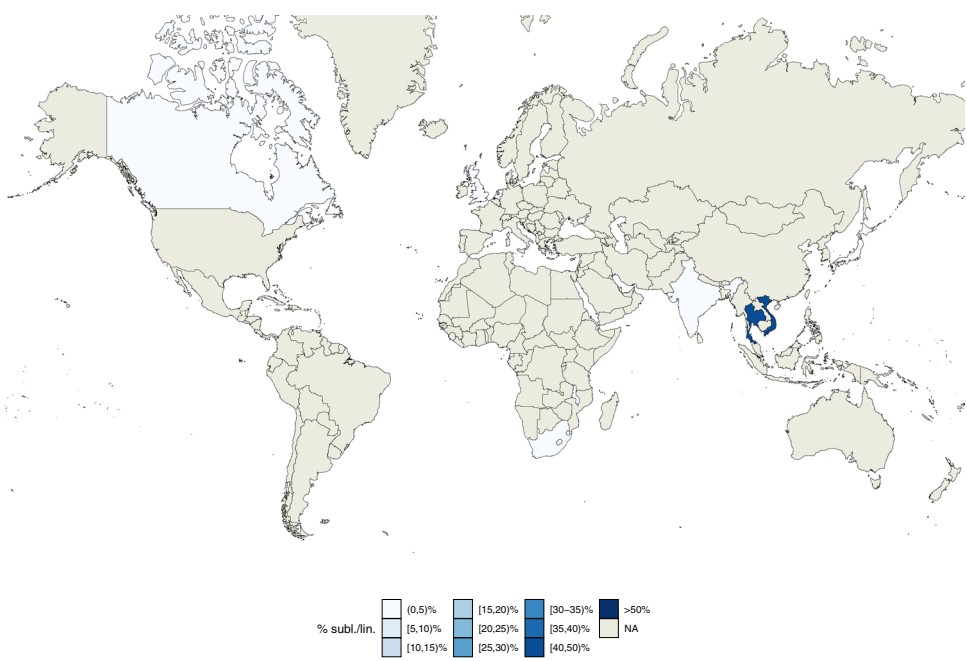

**Fig. 7 Geographic distribution of internal sub-lineage 1.1.1.1.** Colors represent the percentage of 1.1.1.1 strains isolated in a given country with respect to all lineage 1 strains isolated in such country. Source data are provided as a Source Data file.

continents2 (Supplementary Figs. 15 and 16) supporting the hypothesis that L1.1.2 is a geographically unrestricted L1 sub-lineage. The candidate geographically restricted group 1.1.3.i1 from Malawi, was only additionally found in South Africa (Supplementary Fig. 16). L2 isolates were found in all five countries with 98% belonging to the modern-Beijing group (Supplementary Figs. 15 and 17). The modern-Beijing internal group 2.2.1.1.1.i3, corresponding to the Central Asia group[7], was the most prevalent L2 group in Ukraine and Azerbaijan and the second most prevalent group in Pakistan. This confirms 2.2.1.1.1.i3's observed high Sdi (0.66) in the convenience training data and is in line with modern-Beijing's transmissibility (next paragraph and ref. [21]). L3 isolates were found in four (Azerbaijan, Bangladesh, Pakistan, South Africa) of the five countries (Supplementary Fig. 15). Sub-lineage 3.1.1 was the most prevalent sub-lineage (Supplementary Fig. 18), but the two new L3 sub-lineages we describe (3.2.1 and 3.1.2) were also observed at low frequency (2.4-4.8%) in Bangladesh and Pakistan. L4 isolates and most commonly 4.1, 4.3, and 4.10 (that correspond to 4.1, 4.3/ LAM, and 4.10/PGG in the Stucki et al.[8] classification; Supplementary Fig. 19) were found in all five countries in line with their Sdi >0.67 and results on L4 transmissibility below.

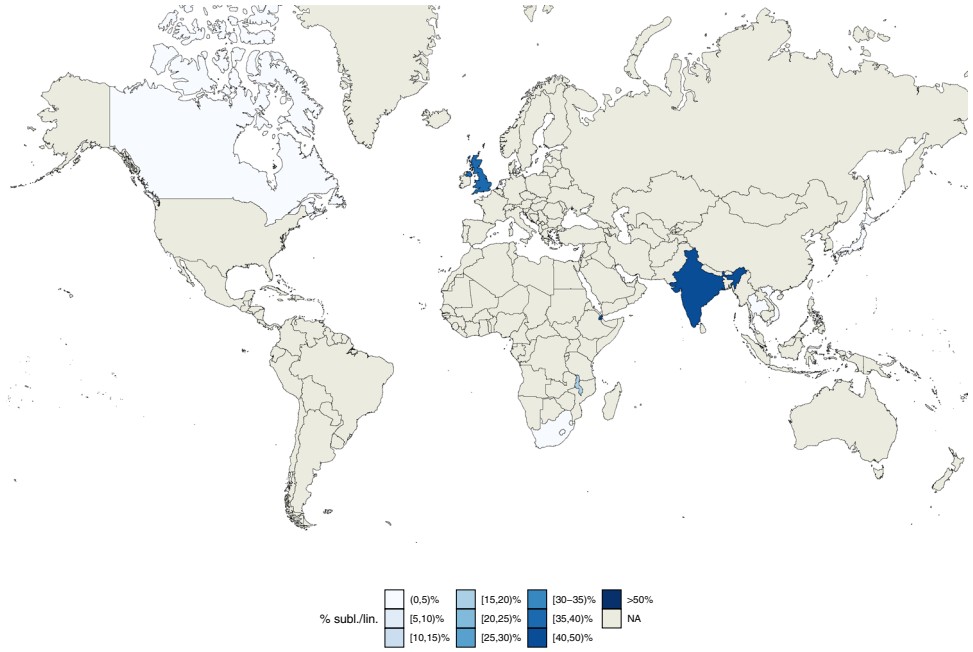

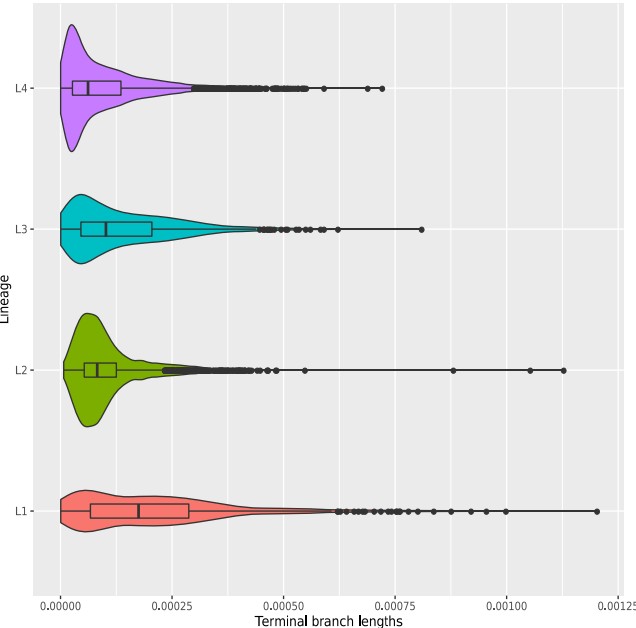

**Fig. 8 Geographic distribution of internal sub-lineage 1.1.2.** Colors represent the percentage of 1.1.2 strains isolated in a given country with respect to all lineage 1 strains isolated in such country. Source data are provided as a Source Data file.

**Differences in transmissibility between the *Mtb* global lineages**. The observation that some lineages/sub-lineages are more geographically widespread than others raises the question of whether this results from differences in marginal transmissibility across human populations. On a topological level, we observed L2 and L3 phylogenies to be qualitatively different from those of L1 and L4 (Figs. 1–4): displaying a star-like pattern with shorter internal branches and longer branches near the termini. We confirmed this quantitatively by generating a single phylogenetic tree for all 9584 L1–4 isolates and plotting cumulative branch lengths from root to tip for each main lineage (Supplementary Fig. 20). Star-like topologies have been postulated to associate with rapid or effective viral or bacterial transmission e.g., a "super-spreading" event in outbreak contexts[25]. To compare transmissibility between the four lineages, we compared the distributions of terminal branch lengths expecting a skew toward shorter terminal branch lengths supporting the idea of higher transmissibility. We found L4 to have the shortest median terminal branch length, followed in order by L2, L3, and L1 (medians: $6.2 \times 10^{-5}$, $8.2 \times 10^{-5}$, $10.2 \times 10^{-5}$, $17.5 \times 10^{-5}$, respectively; all pairwise two-sided Wilcoxon rank-sum tests significant *p*-value < 0.001; Fig. 9). Shorter internal node-to-tip distance is a second phylogenetic correlate of transmissibility; the distribution of this measure across the four lineages revealed a similar hierarchy to the terminal branch length distribution (Supplementary Fig. 21). We also computed the cumulative distribution of isolates separated by increasing total pairwise SNS distance (Supplementary Fig. 22). The proportion of L4 isolates separated by <5 SNS was highest and followed by L2, L3, and L1, respectively. Thus, despite the topological similarities between the L3 and L2 trees, L3 was measured to be less transmissible by interrogating the topology of the trees.

To confirm that the measured transmissibility differences are not due to sampling bias in the source data, we compared the distributions of terminal branch lengths for the four major lineages using the susceptible isolates only ($n = 4939/9584$; data set with curated phenotypes) and using the Zignol et al. data set, where the isolates have been randomly sampled in five countries.

**Fig. 9 Distributions of terminal branch lengths for the four global Mtb lineages (L1–L4).** Two-sided Wilcoxon rank sum tests were performed to test that two distributions were significantly different. Medians: $6.2 \times 10^{-5}$ (L4), $8.2 \times 10^{-5}$ (L2), $10.2 \times 10^{-5}$ (L3), $17.5 \times 10^{-5}$ (L1). Comparisons: L1 vs L2, L3 or L4 (*p*-value < $2.2 \times 10^{-16}$); L2 vs L3 (*p*-value = $3.6 \times 10^{-6}$), L2 vs L4 (*p*-value < $2.2 \times 10^{-16}$); L3 vs L4 (*p*-value < $2.2 \times 10^{-16}$). Description of the distributions (L1: $n = 739$, Min: $0.5 \times 10^{-5}$, 1st Quartile: $6.7 \times 10^{-5}$, Median: $17.5 \times 10^{-5}$, 3rd Quartile: $28 \times 10^{-5}$, Max: $120 \times 10^{-5}$; L2: $n = 2193$, Min: $0.7 \times 10^{-5}$, 1st Quartile: $5.3 \times 10^{-5}$, Median: $8.2 \times 10^{-5}$, 3rd Quartile: $12 \times 10^{-5}$, Max: $110 \times 10^{-5}$; L3: $n = 1103$, Min: $0.5 \times 10^{-5}$, 1st Quartile: $4.5 \times 10^{-5}$, Median: $10.2 \times 10^{-5}$, 3rd Quartile: $20 \times 10^{-5}$, Max: $80 \times 10^{-5}$; L4: n = 5514, Min: $0.2 \times 10^{-5}$, 1st Quartile: $2.6 \times 10^{-5}$, Median: $6.2 \times 10^{-5}$, 3rd Quartile: $13 \times 10^{-5}$, Max: $70 \times 10^{-5}$). Source data are provided as a Source Data file.

In both cases, we found that L4 and L2 have the shortest median terminal branch lengths and L1 the longest. In the Zignol et al. data set, we also found that L4 and L2 terminal branch lengths were shorter than L3's (Supplementary Information, Supplementary Figs. 23 and 24).

## Discussion

Our results provide the most detailed picture of *Mtb*'s population structure to date and extend previous schemas[6,8], particularly for L1 and L3. We describe 7 and 4 new sub-lineages/internal groups, respectively, for these two lineages. The largest improvement was in increasing the resolution of sub-lineage 1.2 (5 subdivisions instead of 2), and in partitioning L3 into 4 groups since previously only a single group with one sub-lineage was characterized (3.1) for L3. We found an internal sub-lineage of 1.1 (1.1.3.i1) that was found almost exclusively in Malawi (85/91 isolates) with nearest neighbors isolated from India. The tMRCA of this group dates back to a point in time between c1497 and c1754. Recent work examining the evolutionary history of L1 concluded that its origin was most likely in South Asia with a tMRCA estimated in the twelfth century AD[16]. Dissemination of L1 out of South Asia may have been related to increase in maritime trade between the continents in this era including seasonal trade following the monsoon season between South Asia and East/Southern Africa. In this study, a group of isolates belonging to sub-lineage 1.1.3 was defined, the vast majority of which are from Malawi, consistent with our results. European contact with the autochthonous populations in South Eastern Africa is estimated to have taken place in the late 15th century, around the time of origin of sub-lineage 1.1.3.i1. Despite the opportunity for dissemination mediated by trade and colonization into Europe and other continents, we observe an unusual pattern of geographic restriction of this group of isolates, consistent with a specialist phenotype. This supports the idea that this is a candidate lineage with adaptation to a specific human genetic background in this region of Africa through co-evolution. This observation can be confirmed as more extensive and systematic pathogen whole genome sequencing becomes available from Sub-saharan Africa.

For L2, we find limited internal diversity with the exception of major splits between the proto-, ancient and modern-Beijing groups. Our schema supports 3 out of 4 groups previously described as either proto-Beijing or ancient-Beijing; the fourth unsupported group in our schema is Asia ancestral 3 previously proposed as a subdivision of ancient Beijing[7]. We also identify a separate new sub-lineage of the ancient-Beijing group (2.2.1.2). Our results show no evidence for sub-lineages inside the modern-Beijing group, but support three internal groups (2.2.1.1.1.i1, 2.2.1.1.1.i2, 2.2.1.1.1.i3) one of which (Central Asia) was previously described by Shitikov et al. Our analysis supports the existing schemas for L4[6,8] and extends them by defining 11 new sub-lineages (and 5 internal groups). In particular, we improve the resolution of known sub-lineages, for instance, 4.2, 4.6.1, and 4.6.2, and define three new sub-lineages (4.11–4.13) which increase the total number of L4 major sub-lineages from 7 (4.1–4.6 and 4.10 in the Stucki et al. classification) to 10.

We define a SNS barcode (95 SNS) that allows the rapid assignment of *Mtb* sub-lineage designations with detailed semantic and hierarchical sub-lineage naming schema from genomic data (.vcf files)[6–8]. Along with this, we provide a software package and comparative tables/figures to facilitate the interchange between five SNS schemas and three *Mtb* sub-lineage naming systems[6–8,26]. We expect these tools to facilitate the use of the expanded *Mtb* classification by scholars of *Mtb* evolution and public health practitioners alike for applications such as the rapid

assessment of potential outbreaks and isolate triage for detailed phylogenetic analyses. Further, low-level susceptibility to specific *Mtb* drugs has been reported for some lineages[27,28]. Improving the resolution of classification at the sub-lineage level can allow for a better understanding of these findings and potentially a more refined prediction of drug resistance profiles.

We find evidence supporting the *Mtb*-human co-evolution hypothesis and its corollary of lineage differential adaptation[8], including a spectrum of transmissibility across the four major *Mtb* lineages. We characterize L4 and L2 as the most transmissible, L1 as the least transmissible one, and L3 showing an intermediate level of transmissibility using different phylogenetic metrics. This is consistent with previous studies that have identified L2 sub-lineages as more transmissible than L1 in Vietnam[21] and Malawi[22]. This result also supports several reports which have failed to identify differences in transmissibility between L2 and L4[29–31]. In order to test the robustness of our findings and minimize the potential sampling bias, we also compared the transmissibility of the major Mtb lineages using a the pan-susceptible isolates form our data set with curated phenotypes ($n = 4939/9584$) and using a data set where isolates were randomly sampled in five countries and we got similar results. Finally, our results are also in line with a larger number of geographically unrestricted "generalist" L4 and L2 sub-lineages observed (compared with L3 or L1).

We find two new candidates with geographically restricted sub-lineages/internal groups, one of them from Malawi and one from South-East Asia (Vietnam and Thailand). We also report evidence of a geographically unrestricted "generalist" sub-lineage in L1 (1.1.2). L1 is the most ancestral of the four main *Mtb* lineages (L1–4), and given the evidence of low transmissibility on average across countries and sub-lineages, the finding of a potential generalist L1 sub-lineage requires further study and validation with additional data.

Our results raise the question of whether the geographically restricted sub-lineages represent cases of specialization or adaptation to specific human populations and whether the geographically unrestricted sub-lineages might have genetic features that allow them to spread efficiently in many different human populations. Although the phylogenies and validated geographic distribution are suggestive of differential adaptation, confirmation of this observation requires control for TB exposure and differences in contact networks using epidemiological data. A limitation of our analysis is the lack of this data and other patient and country-level data, including human ancestry, recent migration, and HIV prevalence. Nevertheless, we find large differences in the geographic distribution of more than 20 *Mtb* sub-lineages across 51 countries and validated using systematically sampled data from these respective geographies. In the future, as additional patient metadata and *Mtb* sequence data become available, follow-up analyses can confirm the geographic distribution of these sub-lineages. In summary, this work provides new insights into the population structure, biogeography, and transmissibility of *Mtb* and demonstrates the use of this information to classify *Mtb* strains and investigate the relationships between population structure, pathogenicity, and transmission.

## Methods

**Phenotypic and geographic location data**. We compiled a first data set of antibiotic susceptibility and resistance data for 11,349 *Mtb* isolates using public databases (Patric)[32] and literature curation[9,33–44]. A summary table of the data and the scripts used to generate it are available at https://github.com/farhat-lab/resdata (v1.0). The isolate metadata including the country of isolation were downloaded using SRAtools v2.9.1 (https://github.com/ncbi/sra-tools). Isolates were filtered according to the procedures described below, resulting in a data set of 9584 isolates. The full list of isolates and countries of isolation are available on Supplementary

Data 1. We used this data set to determine the sub-lineages definitions and study the differences in transmissibility between the major *Mtb* lineages. In order to determine the geographic distribution of the *Mtb* sub-lineages we used a second data set of 17,431 *Mtb* isolates. The full list of isolates and countries of isolation are available on Supplementary Data 4. Finally, in order to validate the geographic distribution of sub-lineages we used a third data set of 3791 *Mtb* isolates from five countries and three continents (Azerbaijan, Bangladesh, Pakistan, South Africa, and Ukraine, Supplementary Data 6). These isolates come from epidemiological surveys designed to be representative of the entire population of TB patients of each one of these countries[18]. A summary table containing the breakdown by country and lineage of all these three datasets is available on Supplementary Data 7.

**Sequencing data analysis and lineage calling**. Sequence read data for the isolates were downloaded from NCBI (the list of BioSamples is available on Supplementary Data 1). We used an implementation of the pipeline proposed by Ezewudo et al.[45] to get the genetic variants that characterize any strain with respect to the *Mtb* reference strain H37Rv. For each isolate, identified by a BioSample, we downloaded all the associated Illumina sequencing runs and we ran PRINSEQ v0.20.4[46] to trim and filter the reads (using an average phred score threshold of 20). Then we ran Kraken v0.10.6[47] and discarded the isolates where <90% of the reads were assigned to *Mtb*. For this purpose, we set up a custom Kraken database, to reduce the memory requirements of the default database (Reference sequences: NC_009565.1 / L4, NC_000962.3 / L4, NC_017524.1 / L4, NC_002755.2 / L4, NC_021054.1 / L2).

Reads were then aligned to the H37Rv (NC_000962.3) reference genome using "BWA mem" v0.7.17[48]. Duplicate reads were removed with Picard v2.9.2 (http://broadinstitute.github.io/picard/). As an additional quality check the coverage of the *Mtb* genome was evaluated using samtools v1.9[49] and all isolates having a coverage of less than 95% of the *Mtb* genome with a depth of at least 10X were dropped. Genetic variants were called using Pilon v1.22[50]. Lineage calls were made using the fast-lineage-caller v1.0 (https://github.com/farhat-lab/fast-lineage-caller), using the Coll et al.[6] and Shitikov et al.[7] SNS schemes.

**Phylogenetic trees**. In order to reduce the probability of having mixed isolates (e.g., isolates from different lineages/sub-lineages that infected the same patient), that could affect the determination of the tree of a given *Mtb* lineage, we computed the F2 lineage-mixture metric[51] and excluded the isolates that had scores greater than 0.5. We separately determined the phylogenetic trees for the pan-susceptible isolates (isolates that were susceptible to both isoniazid and rifampicin as well as all other antibiotics they were tested) and the resistant ones (isolates that were resistant to one or more antibiotics). To generate the trees we merged the .vcf files of the isolates of a given lineage with bcftools v1.9[52]. Then we removed the repetitive, antibiotic resistance and low coverage regions (Supplementary Data 8). We generated a multi-sequence.fasta alignment from the merged.vcf with vcf2phylip v1.5 (https://doi.org/10.5281/zenodo.1257057) and we determined the phylogenetic tree with iqtree v1.6.10 (automatic model selection, 1000 bootstraps).

**$F_{ST}$ and PCA**. The $F_{ST}$ was calculated using the PopGenome R package v2.6.1[53]. PCA analysis was performed using the ade4 R package[54] v1.7-8 (R v3.5.1).

**Sub-lineage definition**. We defined the sub-lineages by evaluating: (1) the information of the tree topology, (2) the support (bootstrap), (3) the $F_{ST}$, and (4) the results of a PCA analysis performed at each node. We considered a node as an informative one if: the support was >95 (iqtree ultrafast bootstrap), the $F_{ST}$ was >0.33 and in the PCA analysis, we were able to identify two groups (since our trees were binary trees). The threshold of the $F_{ST}$ (0.33), we choose was already used to characterize sub-lineages in the previous works[8] and the distributions of the $F_{ST}$ calculated at each internal node of all lineages suggest it is a reasonable value to discriminate between *Mtb* sub-lineages (Supplementary Fig. 25). In a few situations (n = 3) we relaxed one of these constraints either because the node was the first split within a major lineage (split between 1.1 and 1.2, split between 4.1 and 4.[2–6,10–13]) or the group was already identified by previous studies and the results taken together were indicating that the group is valid (4.3.i1). The scripts used to define the sub-lineages/internal groups are available at https://github.com/farhat-lab/mtb-popstruct-2020.

**Hierarchical sub-lineage naming system**. We assigned to all the sub-lineages their major lineage number (1, 2, 3, or 4). For each of the informative nodes in the tree (it is important to notice that we generated binary trees), we assigned a ".1" for the subtree having most of the descendants, while we assigned a ".2" for the subtree having the least descendants (exceptions to this rule were made for 4.1 and 2.1 to retain compatibility with the Coll et al.[6] SNS scheme). We stopped defining groups when a split was not sufficiently supported (ultrafast bootstrap < 95), the $F_{ST}$ was <0.33, or the results of a PCA analysis were not conclusive (see also the Sub-lineage definition section). We also allowed for internal groups, i.e., if one or more ancestors of a given informative node were not informative we considered all the descendants of the informative node as members of an internal group of a given sub-lineage. We, therefore, added a ".i<n>" suffix to design such groups of strains as internal groups (e.g., 4.1.i1).

**Comparison of tree topologies**. In order to compare the topologies of L1–4 trees, we calculated the proportion of tree length as a function of the node level. We used a phylogenetic tree containing all L1–4 isolates. Starting from the MRCA node of each one of the L1–4 sets of isolates we iteratively looked for the children nodes (target nodes) and determined the average patristic distances between the target nodes and the MRCA node. Each one of the iterations represented a node level. We then normalized the resulting averaged distances by the maximum value of the average patristic distance obtained. This algorithm was implemented in R using the ape v5.3[55] and phangorn v2.5.3[56] packages.

**A python lineage caller**. We developed a python module (https://github.com/farhat-lab/fast-lineage-caller) that takes as input a.vcf file and returns the lineage / sub-lineage calls. We provide five SNS schemes for *Mtb*: Coll et al.[6] (all lineages), the Shitikov et al.[7] (L2), Stucki et al.[8] (L4, geographically bounded and unbounded sub-lineages), Lipworth et al.[26] (*Mtb* complex species, *Mtb* main lineages) and the one proposed here (L1–4). Version 1.0 was used for the analyses described in this study.

**Dating the MRCA of internal group 1.1.3.i1**. We calculated the median branch length from the MRCA node of all 1.1.3.i1 isolates and converted it to the number of SNS using the alignment length. We then considered an evolutionary rate bracket of 0.3 to 0.6 substitutions per genome per year and multiplied it for the number of SNS to get a time range corresponding to such branch length.

**Simpson diversity index**. The Simpson diversity index was calculated using the vegan (version 2.5.6; https://cran.r-project.org/package=vegan) R package. The correlations between the Simpson diversity index and the number of continents where a given sub-lineage has been found or the number of isolates of each sub-lineage (Supplementary Figs. 12 and 14) were determined using the *cor* function of the stats[57] (version 3.5.1) R package.

**Reporting summary**. Further information on research design is available in the Nature Research Reporting Summary linked to this article.

## Data availability
A summary table of the first data set of 11,349 public isolates including drug resistance phenotypes and the scripts used to generate it from source databases are available at https://github.com/farhat-lab/resdata (v1.0). Isolates were filtered according to the criteria described in the methods resulting in a data set of 9584 isolates (described in Supplementary Data 1). To determine the geographic distribution of Mtb sub-lineages identified in this data, we used a second data set of 17,431 isolates (accession numbers are provided in Supplementary Data 4). Lastly, to validate the geographic distribution of sub-lineages we used a third data set of 3791 Mtb isolates systematically samples from five countries and three continents (Azerbaijan, Bangladesh, Pakistan, South Africa, and Ukraine and accession numbers are detailed in Supplementary Data 6). Source data are provided with this paper. Any additional data are available from the corresponding authors upon reasonable request. Source data are provided with this paper.

## Code availability
Fast-lineage-caller is available for download at https://github.com/farhat-lab/fast-lineage-caller (v1.0 was used in this study). The scripts used to determine $F_{ST}$ values and that were used in the process of defining sub-lineages/internal groups are available at https://github.com/farhat-lab/mtb-popstruct-2020 (v0.3 was used in this study). Both these software are released under the GPL-3.0 license.

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

## Acknowledgements

We thank Karel Břinda for comments on the manuscript.

## Author contributions

M.R.F. and L.F. conceived the study, designed the analysis and wrote the manuscript. L.F. and R.V.Jr. performed the analyses. A.H., S.M.M.K., A.S., S.T., N.I., A.B., S.N., D.M.C., A.S.D., and M.Z. contributed data. All authors reviewed the manuscript.

## Competing interests

The authors declare no competing interests.
