## [Peer Review File · Nature Communications]

Title: Population structure, biogeography and transmissibility of *Mycobacterium tuberculosis*REVIEWER COMMENTS

Reviewer #1 (Remarks to the Author):

Freschi et al describe a high-resolution analysis of global population structure of Mycobacterium tuberculosis (Mtb) across 4 main lineages, L1-L4. They describe new sub-lineages and 'internal groups' (internal to sub-lineages) within these previously described lineages, hence refining previous Mtb phylogenies. They reveal new geographically restricted lineages that they suggest further support evidence for coevolution between the pathogen and human populations. An augmented set of 95 single nucleotide substitutions (SNS) are proposed that will facilitate designation of M. tuberculosis isolates into their respective lineages/sub-lineages.

The manuscript describes a robust computational approach from which a refined Mtb phylogeny was produced. This extra resolution as compared to the existing phylogenies will be of interest and utility to the TB field. However, I have some comments that I would like to see addressed.

Major comments

1. The naming convention suggested in the manuscript is cumbersome. For example, their new system would replace the current lineage designations 4.3/LAM and 4.10/PGG with 4.2.1.2.1.1 and 4.2.1.1.1.1.1.1; the latter hardly trip off the tongue, while the former designations have the advantage of at least being easier to articulate. Furthermore, long strings of numbers are prone to typographical errors. To ensure wide uptake of a new naming convention by the field and clinical/diagnostic labs, I would urge the authors to rethink their nomenclature. I realise that the authors are following previous convention, but the extra resolution of their system, (with no doubt increased future resolution as more Mtb isolates are sequenced globally), risks a slippery slope of expanded bifurcations leading to unwieldy number strings.
2. The brevity of the discussion does not provide sufficient interpretation or context for the work. For example, the identification of the geographically restricted 1.1.1.2.i1 internal group in Malawi is interesting, but do the authors have any hypothesis as to why the lineage is so localised? At the moment the finding is just stated with only superficial interpretation.
3. In a similar vein, the evidence for geographical restriction of lineages being a function of co- evolution of the pathogen with local human populations is merely repeating an oft stated observation from others. Given that the target journal is Nat Comms, I would like to have seen some attempt at providing more evidence that could substantiate these claims. Instead we are told that "confirmation of this observation requires control for TB exposure and differences in contact networks using epidemiological data". I think we all realise that there is a need for epi metadata to substantiate these claims, which is why it would be good to provide it. Of course, the availability of (good) metadata is one of the major constraints in placing pathogen wgs data in a broader context, but it would have been nice to know whether the authors have tried to obtain such epi data, or their plans to explore this.

Minor

1. As a case in point re by point above, the ease of making a 'number' error is shown in the Introduction where the authors state "We identify and validate 22 novel sub-lineages and 8 additional internal groups, including 6 in L1 and 4 in L3..", while in the Discussion we are told "We describe 7 and 4 new sub-lineages/internal groups, respectively, for (L1 and L3)". Is it 6 or 7 new sub-lineages in L1? I believe it is 6.
2. The lack of line numbers on the manuscript makes it difficult for a reviewer to highlight areas needing correction/comments.
3. There are some typos in the References (e.g. 6, 7, 40)

Reviewer #2 (Remarks to the Author):

In Freschi et al, the authors used multiple datasets of previously published whole genomes of *M. tuberculosis* clinical isolates and performed various analyses to refine the phylogeny of this important pathogen. In particular, this work extends prior understanding of *M. tuberculosis* phylogeny and population structure with respect to sublineages and define additional sub-lineages and internal groups that had previously not been named. They propose a rational nomenclature for *M. tuberculosis* sublineages, and identify a single nucleotide substitution (SNS) that can be used to identify these sublineages for future analyses. Lastly, they performed a terminal branch length analysis of *Mtb* lineages 1-4, and extrapolate differential branch length findings to transmissibility these lineages.

While much of the analyses are descriptive, this paper does modestly improve our current understanding of the TB phylogeny and propose a useful tool for sublineage nomenclature. There are a few clarifications that would enhance this manuscript in its current form.

Major comments:

1. Even in a seemingly large dataset of over 10K *Mtb* isolates, when over 1 million people on the planet are known to have TB disease each year, there are issues with sampling biases that must be considered. This is a particular issue in this study, which utilizes only previously published genomes that have been sequenced for other purposes. While the authors acknowledge that sampling bias is a limitation, it is not clear that sampling bias has been minimized. For example, how was this sampling issue addressed in the distribution of terminal branch length analysis?
2. Inclusion of a summary table of the *M. tuberculosis* isolates included in each analysis of this study would be helpful to understand the total numbers of isolates from each prior investigation, from each country, the distribution of drug-resistant strains, and distribution of lineages and sublineages to allow the reader to understand the sampling and potential biases. While some of this information is embedded within supplemental Files 1, 4 and 5, these are currently unwieldy and require the reader to tabulate summary numbers of interest.

Minor comments:

1. It would be helpful to include a table of the new sublineage naming system to prior naming systems to ease the comparison with the published literature
2. The conclusion that L4 is as or more transmissible than L2 raises more questions than answers as this appears to differ from the conclusions of other published studies that are cited herein on the global dominance of the Beijing strains.

Reviewer #3 (Remarks to the Author):

The manuscript reads well and it represents a real advance in the TB molecular epidemiology and phylogeography field.

Important notions are discussed, and newly defined TB sub-lineages are proposed.

However, I have a few minor remarks:

On page 5 (line 166) Please be more precise about the "species other than Mtb".

It would be great to provide geographic (or country) distribution for each identified SNS TB sub-lineage. It is not clear how the "fast-lineage-caller" has been validated and compared to other tools. Could you please provide a supplemental table for it?

On page 8 (line 290) "... one from South-East Asia" (please be more precise on the location).

Page 9, lines 336-337, could you please provide the corresponding lineage for each NCBI Reference sequence (NC...../L1, NC...../L2, ...)

Furthermore, regarding data on L1 and L3, it would be interesting to mention recent studies on the subject (<https://doi.org/10.1101/2020.10.20.346866> , <https://doi.org/10.1371/journal.pone.0219706> , <https://doi.org/10.1111/mec.15120>)

**Response to Referees (NCOMMS-20-41049A)**

We thank the reviewers for their constructive comments and suggestions. Here we address one
by one all their comments and provide our answers.

Reviewer #1 (Remarks to the Author):

Freschi et al describe a high-resolution analysis of global population structure of Mycobacterium
tuberculosis (Mtb) across 4 main lineages, L1-L4. They describe new sub-lineages and 'internal
groups' (internal to sub-lineages) within these previously described lineages, hence refining
previous Mtb phylogenies. They reveal new geographically restricted lineages that they suggest
further support evidence for coevolution between the pathogen and human populations. An
augmented set of 95 single nucleotide substitutions (SNS) are proposed that will facilitate
designation of M. tuberculosis isolates into their respective lineages/sub-lineages.

The manuscript describes a robust computational approach from which a refined Mtb phylogeny
was produced. This extra resolution as compared to the existing phylogenies will be of interest
and utility to the TB field. However, I have some comments that I would like to see addressed.

Major comments

1. The naming convention suggested in the manuscript is cumbersome. For example, their new
system would replace the current lineage designations 4.3/LAM and 4.10/PGG with 4.2.1.2.1.1
and 4.2.1.1.1.1.1.1; the latter hardly trip off the tongue, while the former designations have the
advantage of at least being easier to articulate. Furthermore, long strings of numbers are prone
to typographical errors. To ensure wide uptake of a new naming convention by the field and
clinical/diagnostic labs, I would urge the authors to rethink their nomenclature. I realise that the
authors are following previous convention, but the extra resolution of their system, (with no doubt
increased future resolution as more Mtb isolates are sequenced globally), risks a slippery slope
of expanded bifurcations leading to unwieldy number strings.

Author's Response:

In our work we propose a hierarchical naming system for Mtb, which has the advantages of
directly communicating the phylogenetic relationships between groups, and automating the
process of lineage classification when new isolates are considered. The hierarchical naming
scheme does have, as the reviewer points out, the disadvantage of long names (in particular for
L4) due to the complex phylogeny. We acknowledge that long names may be difficult to remember
by human scientists, and prone to errors if handled manually.

We believe that the advantages of the hierarchical naming scheme overall outweigh its
disadvantages since:

- - The main naming system currently in use, *i.e.* Coll *et al.*, already contains designations
that describe five subdivisions (e.g. 4.2.1.2.1). Simply extending such a naming system
using our new results, which results in adding new groups, readily leads to designations

with seven subdivisions. In addition, the number of subdivisions is expected to grow with
time, potentially leading to the same issues raised for the hierarchical naming system
(which currently has a maximum of eleven subdivisions).

- People already use spoligotypes, which are long strings of zeros and ones, to type *Mtb*
strains and share them with the community. Spoligotypes are organized in families, to
make it easier for the end users to understand which group the spoligotype designates. In
the case of SNS there could be a shorthand naming system for everyday use and a
systematic one to work with the details.

To address the reviewer's appropriate concern about naming length we now revise the naming to
a new shorthand lineage naming scheme that's based on the Coll *et al.* naming scheme. We
basically extend the Coll *et al.* lineage designations, but try to retain some of the concepts
developed for the hierarchical naming system specifically the internal groups (groups which have
ancestor nodes where the topology of the tree cannot be fully resolved). These lineage
designations are shorter than the hierarchical scheme, in particular for L4 strains. This means
that the group 4.3.3 in Coll *et al.* (a schema that does not distinguish between true sub-lineages
and internal groups) is now named 4.3.i3 as the tree topology in fact does not support one or
more of its ancestor nodes. Also, 4.10 remains with the same designation as Coll *et al.* instead of
4.2.1.1.1.1.1.1, improving the overall readability. For L1-3 isolates the designations are almost
interchangeable with the hierarchical nomenclature, thus allowing quick phylogenetic
comparisons.

We would like to note that we considered other solutions to compress the hierarchical lineage
designations e.g. converting them to hexadecimal, using combinations of letters and numbers to
shorten them, using letters to define the main sub-lineages and then define clusters of isolates
with numbers, but all these attempts implied a loss of information or making the naming system
even more complex to read. We note that the SARS-CoV2 genomic surveillance community has
noted similar challenges to naming strains and clades and have expressed challenges with
appropriate naming that often require long cumbersome names¹.

We now discuss these points in the main text (revised text highlighted):
"To better classify *Mtb* isolates in the context of the global *Mtb* population structure, we developed
a hierarchical sub-lineage naming scheme (Suppl. File 2) [...]. This proposed system overcomes
two major shortcomings of the existing schemas: same-level sub-lineages are never overlapping
(unlike the system of Stucki *et al.*² sub-lineage 4.10 includes sub-lineages 4.7–4.9), and the
names reflect both phylogenetic relationships and genetic similarity (unlike *semantic* naming such
as the "Asia ancestral" lineage in the system of Shitikov *et al.*³). Further, this naming system can
be standardized to automate the process of lineage definition. These advantages come at the
price of long sublineage names in the case of complex phylogenies (e.g. for L4, sub-lineage 4.10
gets the lineage designation 4.2.1.1.1.1.1.1). For compatibility with naming conventions already
in use and to keep names as short as possible, we designed a second, shorthand, naming system
which expands the Coll *et al.* lineage schema by adding new subdivisions and differentiating
between sub-lineages and internal groups. For instance, sublineage 4.3.1 is designated as 4.3.i1,
informing the user that this is an internal group of sublineage 4.3. To simplify the use of the

hierarchical naming schema and the updated shorthand schema, we provide a table that
compares them side by side along with naming systems currently in use (Suppl. File 2).

2. The brevity of the discussion does not provide sufficient interpretation or context for the work.
For example, the identification of the geographically restricted 1.1.1.2.i1 internal group in Malawi
is interesting, but do the authors have any hypothesis as to why the lineage is so localised? At
the moment the finding is just stated with only superficial interpretation.

Author' s Response:

In response to the reviewers comment we have now substantially revised the discussion around
the identification of the internal group in Malawi. We performed an approximate molecular dating
and drew more on the literature for the context of this group and its origins. This is described in
the quoted text of the manuscript below. We also added a new **Supplementary Figure 3** to
show in detail the phylogenetic context of sub-lineage 1.1.1.2.i1 (now named 1.1.3.i1 in the
shorthand designation).

Suppl. Figure 3. Phylogenetic context of the internal sub-lineage 1.1.3.i1 / Malawi.

We have also added the following text to the results section:

We [...] detected an internal group of 91 isolates (1.1.3.i1) characterized by a long defining branch
in the phylogeny (corresponding to 82 SNSs), a high F_{ST} (0.48), and geographically restricted to
Malawi (85/91, 93% isolates, Fig. 1 and Suppl. Fig. 3). We approximated the time to the most
recent common ancestor (tMRCA, Methods) of this group at c1497 to 1754.

The following text has been added to the discussion:
We found an internal sub-lineage of 1.1 (1.1.3.i1) that was found almost exclusively in Malawi
(85/91 isolates) with nearest neighbors isolated from India. The tMRCA of this group dates back
to a point in time between c1497 and c1754. Recent work examining the evolutionary history of
L1 concluded that its origin was most likely in South Asia with a tMRCA estimated in the 12th
century AD⁴. Dissemination of L1 out of South Asia may have been related to increase in maritime
trade between the continents in this era including seasonal trade following the monsoon season
between South Asia and East/Southern Africa. In this study, a group of isolates belonging to sub-
lineage 1.1.3 was defined, the vast majority of which are from Malawi, consistent with our results.
European contact with the autochthonous populations in South Eastern Africa is estimated to
have taken place in the late 15th century, around the time of origin of sublineage 1.1.3.i1. Despite
the opportunity for dissemination mediated by trade and colonization into Europe and other
continents we observe an unusual pattern of geographic restriction of this group of isolates,
consistent with a specialist phenotype. This supports the idea that this is a candidate lineage with
adaptation to a specific human genetic background in this region of Africa through co-evolution.
This observation can be confirmed as more extensive and systematic pathogen whole genome
sequencing becomes available from Sub-Saharan Africa.

3. In a similar vein, the evidence for geographical restriction of lineages being a function of co-
evolution of the pathogen with local human populations is merely repeating an oft stated
observation from others. Given that the target journal is Nat Comms, I would like to have seen
some attempt at providing more evidence that could substantiate these claims. Instead we are
told that “confirmation of this observation requires control for TB exposure and differences in
contact networks using epidemiological data”. I think we all realise that there is a need for epi
metadata to substantiate these claims, which is why it would be good to provide it. Of course, the
availability of (good) metadata is one of the major constraints in placing pathogen wgs data in a
broader context, but it would have been nice to know whether the authors have tried to obtain
such epi data, or their plans to explore this.

Author's Response:
We thank the reviewer for this comment. To our knowledge the prior published work on
geographic restriction of TB lineage is sparse and limited to Lineage 4. This is the first time that
this notion is evaluated at this scale across the Mtb phylogeny. We have adapted new metrics to
quantify geographic restriction and criteria to do so, and we show that there is only a weak
correlation the Simpson index and the number of isolates ($\rho = 0.34$, $p\text{-value} = 0.03$), suggesting
that specialists are not observed only due to lower sampling.

Although we don't have data on TB exposure of the hosts from which the Mtb samples were
isolated, we do validate the geographic distribution of sub-lineages among samples collected
systematically for surveillance by the World Health Organization across five countries. This is the
first time that such systematically collected isolates are used to assess phylogeography of Mtb,
previous studies only focused on convenience samples.

We explored the possibility of gathering data on exposure and other epidemiological parameters
to link to the whole genome sequences but this would involve a timeline of years which is not
compatible with reasonable publication times. However, this project has prompted us to go
towards the direction pointed out by the reviewer and we started a new project in which we will
gather both genome sequence data and epidemiological data (Pending NIH/NIAID R21
AI154089-01A1) to study how diversity in mycobacterial genes could mediate adaptation to
humans of different ancestry or their environments.

Minor

1. As a case in point re by point above, the ease of making a 'number' error is shown in the
Introduction where the authors state "We identify and validate 22 novel sub-lineages and 8
additional internal groups, including 6 in L1 and 4 in L3..", while in the Discussion we are told "We
describe 7 and 4 new sub-lineages/internal groups, respectively, for (L1 and L3)". Is it 6 or 7 new
sub-lineages in L1? I believe it is 6.

Authors Response:

The confusion here is due to the definitions of sub-lineage and internal group. We find 6 new sub-
lineages in L1, plus an internal group (1.1.3.i1), bringing the total number of new L1 groups
defined here to 7. In the introduction section our intent was to report the number of sub-lineages,
but it is not clear from our sentence and we apologize for this. To make the text more uniform, we
listed the total number of groups in the introduction as we do in the discussion:

"We identify and validate 22 novel sub-lineages and 8 additional internal groups (*i.e.* genetically
divergent groups found in sub-lineages that cannot be further partitioned in a hierarchical fashion
according to our criteria), including 7 in L1 and 4 in L3, and expand the SNS typing barcode to 95
sites".

2. The lack of line numbers on the manuscript makes it difficult for a reviewer to highlight areas
needing correction/comments.

Author's Response: We sincerely apologize regarding this omission. We fixed this issue and
added the line numbers to all files.

3. There are some typos in the References (e.g. 6, 7, 40)

Author's Response: We corrected the typos. Reference #40 (now 49) is a reference to an arXiv
pre-print, which is the most appropriate according to the authors of the software (see
<https://github.com/lh3/bwa>).

Reviewer #2 (Remarks to the Author):

In Freschi et al, the authors used multiple datasets of previously published whole genomes of M.
tuberculosis clinical isolates and performed various analyses to refine the phylogeny of this
important pathogen. In particular, this work extends prior understanding of M. tuberculosis

phylogeny and population structure with respect to sublineages and define additional sub-
lineages and internal groups that had previously not been named. They propose a rational
nomenclature for *M. tuberculosis* sublineages, and identify a single nucleotide substitution (SNS)
that can be used to identify these sublineages for future analyses. Lastly, they performed a
terminal branch length analysis of *Mtb* lineages 1-4, and extrapolate differential branch length
findings to transmissibility these lineages.

While much of the analyses are descriptive, this paper does modestly improve our current
understanding of the TB phylogeny and propose a useful tool for sublineage nomenclature. There
are a few clarifications that would enhance this manuscript in its current form.

Major comments:

1. Even in a seemingly large dataset of over 10K *Mtb* isolates, when over 1 million people on
the planet are known to have TB disease each year, there are issues with sampling biases that
must be considered. This is a particular issue in this study, which utilizes only previously
published genomes that have been sequenced for other purposes. While the authors
acknowledge that sampling bias is a limitation, it is not clear that sampling bias has been
minimized. For example, how was this sampling issue addressed in the distribution of terminal
branch length analysis?

Authors Response:

We thank the reviewer for raising this point. Minimizing sampling bias was a key scientific priority
for us as we conducted the analysis. To achieve this we relied on the largest dataset of isolates
available to us. For the phylogenetic analysis and lineage definitions we specifically collected
isolates that had known antibiotic resistance phenotypes as a major source of biased sampling is
over-representation of resistance in published datasets. Accordingly we stratified the analysis by
resistance phenotype, assessing sub-lineages only among susceptible isolates and reserving the
phylogeny of resistant isolates for validation. We also specifically include a third dataset
systematically collected for surveillance purposes by five countries under the guidance of the
World Health Organization ⁵.

We also pooled data across geographies for inference regarding relative transmissibility of the 4
major lineages. Hence our measure of transmissibility is relative between lineages and averaged
over countries. This averaging and relative measurement of transmissibility minimizes the effect
of bias as there is no reason that sampling bias will preferentially affect *Mtb* isolates from a specific
lineage that is only known after sequencing.

We further used three metrics (terminal branch lengths, node-to-tip distances for all internal
nodes, proportions of isolates belonging to a given lineage as a function of pairwise SNS
difference) to look at transmissibility and all the three metrics show similar results. Finally, when
stating our conclusions we were careful to not over step our interpretation of the results and we
focus on the largest differences i.e. between L1 and (L2 or L4) and between L3 and (L2 or L4).
The phylogenetic differences between (L2 or L4) and (L1 or L3) are highly statistically significant
with P-values < 3.6 ×10⁻⁶.

In order to further confirm that the measured differences in transmissibility are not due to sampling
bias we performed two new analyses: (1) we compared the terminal branch lengths distributions
of susceptible isolates only, belonging to the major Mtb lineages (4,939 isolates), This would
eliminate any oversampling bias of drug resistance. And (2) we compared the terminal branch
lengths distributions between the major Mtb lineages using the WHO dataset, where the isolates
have been systematically sampled to reliably represent the entire population of TB patients in five
countries. We find that the order of transmissibility holds in these validation analyses with L2 or
L4 being more transmissible than L3 and L1 respectively.

We report describe these results in the Results section with more details in the Supplement
section:

Results text added:

“To confirm that the measured transmissibility differences are not due to sampling bias in the
source data, we compared the distributions of terminal branch lengths for the four major lineages
using the susceptible isolates only (n = 4,939/9,584; dataset with curated phenotypes) and using
the Zignol *et al.* dataset, where the isolates have been randomly sampled in five countries. In both
cases we found that L4 and L2 have the shortest median terminal branch lengths and L1 the
longest. In the Zignol *et al.* dataset we also found that L4 and L2 terminal branch lengths were
shorter than L3’s (Supplementary information, Suppl. Fig. 23-24).”

Supplement text added:

“To confirm that the measured transmissibility differences are not due to sampling bias in the
source data, we compared the distributions of terminal branch lengths for the four major lineages
using the susceptible isolates only (n = 4,939/9,584; dataset with curated phenotypes). We found
L4 to have the shortest median terminal branch length and L1 the longest (median terminal branch
length: L4=8.7×10⁻⁵, L2=10.1×10⁻⁵, L3=9.8×10⁻⁵, L1=17×10⁻⁵); two sided pairwise Wilcoxon rank
sum tests were significant at the multiple testing corrected threshold of P-value<0.001 except
between L2 vs L3, P-value = 0.05; Suppl. Fig. 23. The lack of significant difference between L2
and L3 in this drug susceptible dataset is likely due to smaller sample size. The fact that L2 is
recognized to have a higher rate of drug resistance than other lineages resulted in more filtering
of L2 isolates when we restricted to the drug susceptible subset ⁶. We also compared the
distributions of terminal branch lengths between the four major lineages using the Zignol *et al.*
dataset, where the isolates have been randomly sampled in five countries. In this case we again
found that L4 and L2 have the shortest median terminal branch length and L1 the longest (median
terminal branch length: L2=11.8×10⁻⁵, L4=16.1×10⁻⁵, L3=29.3×10⁻⁵, L1=37×10⁻⁵, respectively; all
pairwise two sided Wilcoxon rank sum tests significant with P-values<0.001; Suppl. Fig. 24).”

2. Inclusion of a summary table of the M. tuberculosis isolates included in each analysis of this
study would be helpful to understand the total numbers of isolates from each prior investigation,
from each country, the distribution of drug-resistant strains, and distribution of lineages and
sublineages to allow the reader to understand the sampling and potential biases. While some of

this information is embedded within supplemental Files 1, 4 and 5, these are currently unwieldy
and require the reader to tabulate summary numbers of interest.

**Authors Response:**

We now include a Supplementary File (Suppl. File 7) with summary tables that show the
distribution of the isolates by country, main lineage and sub-lineage. The distribution of the
isolates. Here are two excerpt from these tables as the full tables are too long to display here.:

**Sheet 1 / Excerpt from distribution of the isolates by country for the three datasets used in this**
**work.**

country	9K_with_phenotypes	ZIGNOL	NCBI
#	1158	0	0
Albania	0	0	9
Argentina	0	0	169
Australia	0	0	77
Azerbaijan	1	707	135
Bangladesh	0	635	25
Belarus	136	0	37

**Sheet 2 / Excerpt from distribution of the isolates by sub-lineages for the three datasets used in**
**this work.**

lineage	9K_with_phenotypes	NCBI	ZIGNOL
1.1	0	2	0
1.1.1	0	12	0

1.1.1.1	33	362	2
1.1.1.2	4	4	0
1.1.2	264	277	38
1.1.3	49	71	240
1.1.3.i1	91	107	1

Minor comments:

1. It would be helpful to include a table of the new sublineage naming system to prior naming
systems to ease the comparison with the published literature

Authors Response:

We have now added two more tables (Excel sheets) to Suppl. File 2 to facilitate the comparison
between the different naming systems (“cmp_schemes” and “cmp_schemes_all_nodes”). The
former reports the designations for recent sub-lineages (i.e. the subdivisions that provide the
highest resolution in our naming scheme and are the closest ones to the tips of the tree; e.g.
1.2.2.1) in the different naming schemes; the latter reports the designations of all sub-lineages in
the different naming systems, meaning that it will list, for instance, the designations for 1, 1.2,
1.2.2 and 1.2.2.1. The updated lineage shorthand, lineage_hierarchical and two previously
published schema (Coll et al and Shitikov et al) are compared.

Here is an excerpt of the table “cmp_schemes”:

lineage	lineage_hierarchical	coll	shitikov
1.2.2.1	1.2.2.1	1.2.2	NA
...
2.1	2.1	2.1	proto_beijing
...
4.1	4.1	4.1	NA

4.5	4.2.1.1.2	4.5	NA
4.11	4.2.1.1.1.1.2	4	NA
4.10	4.2.1.1.1.1.1.1	4.[7-9]	NA

Here is an extract of the table “cmp_schemes_all_nodes”:

lineage	lineage_hierarchical	coll	shitikov
1	1	1	NA
1.1	1.1	1.1	NA
1.2	1.2	1.2	NA
1.2.2	1.2.2	1.2.2	NA
1.2.1	1.2.1	1.2.1	NA
1.2.2.1	1.2.2.1	1.2.2	NA

2. The conclusion that L4 is as or more transmissible than L2 raises more questions than answers as this appears to be differ from the conclusions of other published studies that are cited herein on the global dominance of the Beijing strains.

Author’s Response

We thank the reviewer for raising this point. We want to note that previous works compared the different Mtb lineages at the local level, meaning that they have a different and partial sub-lineage coverage. We also note that both L2 and L4 are recognized to contain the vast majority of the known Mtb “generalist” sub-lineages, which supports the idea of both these lineages being highly transmissible.

Our conclusions are not in contrast with the previous works, since we concluded that L2 and L4 are more transmissible than L3 and L1, with L1 being the least transmissible Mtb lineage. In our response above on minimizing sampling bias we describe that validation we have conducted to confirm the associations we observe between transmissibility across the 4 major Mtb lineages are

343 robust. Further existing reports in the literature vary on the relative transmissibility of the four
lineages with L1 and L3 being less studied. Several reports have failed to identify differences in
transmissibility between L2 and L4 e.g. PMID: 26224845, PMID: 24849817, PMID: 29422032

Our dataset is largest to date measuring transmissibility across lineages and across
geographies and we believe this adds to the existing local reports on this question.

We have now added more discussion on differential transmissibility of the four lineages to the
discussion section that currently reads:

“We find evidence supporting the *Mtb*-human co-evolution hypothesis and its corollary of lineage
differential adaptation², including a spectrum of transmissibility across the four major *Mtb*
lineages. We characterize L4 and L2 as the most transmissible, L1 as the least transmissible one
and L3 showing an intermediate level of transmissibility using different phylogenetic metrics. This
is consistent with previous studies that have identified L2 sub-lineages as more transmissible than
L1 in Vietnam ⁷ and Malawi ⁸. This result also supports several reports which have failed to
identify differences in transmissibility between L2 and L4 ^{9–11}. In order to test the robustness of
our findings and minimize the potential sampling bias we also compared the transmissibility of the
major *Mtb* lineages using a the pan-susceptible isolates form our dataset with curated phenotypes
(n = 4,939/9,584) and using a dataset where isolates were randomly sampled in five countries
and we got similar results. Finally, our results are also in line with a larger number of
geographically unrestricted ‘generalist’ L4 and L2 sub-lineages observed (compared with L3 or
L1).”

REVIEWER COMMENTS

Reviewer #3 (Remarks to the Author):

The manuscript reads well and it represents a real advance in the TB molecular epidemiology
and phylogeography field.

Important notions are discussed, and newly defined TB sub-lineages are proposed.

Authors’ Response: We thank the reviewer for their kind words and thoughts that have helped us
improve the manuscript.

However, I have a few minor remarks:

On page 5 (line 166) Please be more precise about the "species other than *Mtb*".

Authors’ Response:

We now made clear that with the appropriate barcode fast-lineage-caller can be used for any
microbial species: “The tool is generalizable and can manage additional barcodes defined by the
user to type the core genome of potentially any bacterial species.”

It would be great to provide geographic (or country) distribution for each identified SNS TB sub-
lineage.

**Authors Response:**

We added tables (Suppl. File 7) and maps (Suppl. File 5) that show the world distribution of
each of the sub-lineages/internal groups described in our work.

It is not clear how the "fast-lineage-caller" has been validated and compared to other tools.
Could you please provide a supplemental table for it?

**Authors Response:**

Fast-lineage-caller is a Software package written for Python that allows lineage calling from .vcf
files using different SNP schemes.

It can be downloaded from <https://github.com/farhat-lab/fast-lineage-caller>, and the basic syntax
is the following:

`fast-lineage-caller my_isolate.vcf`

The output is a table of the lineage calls with the different SNP schemes:

Isolate	coll2014	freschi2020	lipworth2019	shitikov2017	stucki2016
SAMEA968141	lineage2.2.1	2.2.1.1.1	beijing	lin2.2.1,asian_african_2	NA

The package can also provide information on how many SNPs support the lineage calls (if an
option is selected):

Isolate	coll2014	freschi2020	lipworth2019	shitikov2017	stucki2016
SAMEA968141	lineage2.2.1(1/1)	2.2.1.1.1(1/1)	beijing(296/296)	lin2.2.1(3/3),asian_african_2(2/2)	NA

The package includes all the most used/up-to-date SNP schemes, while the other available
software mostly include a single SNP scheme. We will continue to add SNP schemes when new
studies are published and add new features as well (i.e. lineage calls from FASTA/Q files, scripts
to import/export lineage schemes from/to other tools, RD and INDEL schemes).

On page 8 (line 290) "... one from South-East Asia" (please be more precise on the location).

Authors Response: we now specify in the text the countries where sub-lineage 1.1.1.1.1 is
found:

“We find two new candidate geographically restricted sub-lineages/internal groups, one of them
from Malawi and one from South-East Asia (Vietnam and Thailand).”

Page 9, lines 336-337, could you please provide the corresponding lineage for each NCBI
Reference sequence (NC...../L1, NC...../L2, ...)

Authors Response: we added the corresponding lineages as suggested by the reviewer:

“For this purpose we set up a custom Kraken database, to reduce the memory requirements of
the default database (Reference sequences: NC_009565.1 / L4, NC_000962.3 / L4,
NC_017524.1 / L4, NC_002755.2 / L4, NC_021054.1 / L2). “

Furthermore, regarding data on L1 and L3, it would be interesting to mention recent studies on
the subject (<https://doi.org/10.1101/2020.10.20.346866> ,
<https://doi.org/10.1371/journal.pone.0219706> , <https://doi.org/10.1111/mec.15120>)

Authors Response:

We thank the reviewer for pointing out these references, we now cite and discuss these studies
in our main text as detailed below. We note that some of these references overlap with new
literature we cited to expand on the Malawi L1.1.3.i1 lineage in response to Reviewer 1’s
comments.

Introduction

[...] L1 and L3 diversity is less understood as these lineages are most prevalent in countries where
pathogen sequencing has been less widely applied, but this is rapidly changing due to the
increasing sequencing capacity in high-burden TB settings and supported by international
research collaborations ^{4,5}. The population structure of L1 and L3 is less understood as these
lineages are most prevalent in countries where pathogen sequencing had been less widely
applied. Recently, studies fueled by increasing sequencing capacity in high-burden TB settings
have begun to evaluate the evolutionary history of L1 and L3 including the role of migration and
dispersal in driving their prevalence in different parts of the world^{4,5,12,13}. [...]

Discussion

We found an internal sub-lineage of 1.1 (1.1.3.i1) that was found almost exclusively in Malawi
(85/91 isolates) with nearest neighbors isolated from India. The tMRCA of this group dates back
to a point in time between c1497 and c1754. Recent work examining the evolutionary history of
L1 concluded that its origin was most likely in South Asia with a tMRCA estimated in the 12th
century AD⁴. Dissemination of L1 out of South Asia may have been related to increase in maritime
trade between the continents in this era including seasonal trade following the monsoon season
between South Asia and East/Southern Africa. In this study, a group of isolates belonging to sub-
lineage 1.1.3 was defined, the vast majority of which are from Malawi, consistent with our results.
European contact with the autochthonous populations in South Eastern Africa is estimated to
have taken place in the late 15th century, around the time of origin of sublineage 1.1.3.i1. Despite

the opportunity for dissemination mediated by trade and colonization into Europe and other
continents we observe an unusual pattern of geographic restriction of this group of isolates,
consistent with a specialist phenotype. This supports the idea that this is a candidate lineage with
adaptation to a specific human genetic background in this region of Africa through co-evolution.
This observation can be confirmed as more extensive and systematic pathogen whole genome
sequencing becomes available from Sub-saharan Africa.

- 1. Callaway, E. 'A bloody mess': Confusion reigns over naming of new COVID variants.
*Nature* vol. 589 339 (2021).
- 2. Stucki, D. *et al.* Mycobacterium tuberculosis lineage 4 comprises globally distributed and
geographically restricted sublineages. *Nat. Genet.* **48**, 1535–1543 (2016).
- 3. Shitikov, E. *et al.* Evolutionary pathway analysis and unified classification of East Asian
lineage of Mycobacterium tuberculosis. *Sci. Rep.* **7**, 9227 (2017).
- 4. Menardo, F. *et al.* Local adaptation in populations of Mycobacterium tuberculosis endemic
to the Indian Ocean Rim. *Cold Spring Harbor Laboratory* 2020.10.20.346866 (2020)
doi:10.1101/2020.10.20.346866.
- 5. Zignol, M. *et al.* Genetic sequencing for surveillance of drug resistance in tuberculosis in
highly endemic countries: a multi-country population-based surveillance study. *Lancet*
*Infect. Dis.* **18**, 675–683 (2018).
- 6. Ektefaie, Y., Dixit, A., Freschi, L. & Farhat, M. R. Globally diverse Mycobacterium
tuberculosis resistance acquisition: a retrospective geographical and temporal analysis of
whole genome sequences. *The Lancet Microbe* **2**, e96–e104 (2021).
- 7. Holt, K. E. *et al.* Frequent transmission of the Mycobacterium tuberculosis Beijing lineage
and positive selection for the EsxW Beijing variant in Vietnam. *Nat. Genet.* **50**, 849–856
(2018).
- 8. Guerra-Assunção, J. A. *et al.* Large-scale whole genome sequencing of M. tuberculosis
provides insights into transmission in a high prevalence area. *Elife* **4**, (2015).
- 9. Nebenzahl-Guimaraes, H., Verhagen, L. M., Borgdorff, M. W. & van Soolingen, D.

- Transmission and Progression to Disease of Mycobacterium tuberculosis Phylogenetic
Lineages in The Netherlands. *J. Clin. Microbiol.* **53**, 3264–3271 (2015).
- 10. Nebenzahl-Guimaraes, H., Borgdorff, M. W., Murray, M. B. & van Soolingen, D. A novel
approach - the propensity to propagate (PTP) method for controlling for host factors in
studying the transmission of Mycobacterium tuberculosis. *PLoS One* **9**, e97816 (2014).
- 11. Peres, R. L. *et al.* Risk factors associated with cluster size of Mycobacterium tuberculosis
(Mtb) of different RFLP lineages in Brazil. *BMC Infect. Dis.* **18**, 71 (2018).
- 12. Couvin, D., Reynaud, Y. & Rastogi, N. Two tales: Worldwide distribution of Central Asian
(CAS) versus ancestral East-African Indian (EAI) lineages of Mycobacterium tuberculosis
underlines a remarkable cleavage for phylogeographical, epidemiological and
demographical characteristics. *PLOS ONE* vol. 14 e0219706 (2019).
- 13. O'Neill, M. B. *et al.* Lineage specific histories of Mycobacterium tuberculosis dispersal in
Africa and Eurasia. *Molecular Ecology* vol. 28 3241–3256 (2019).

REVIEWERS' COMMENTS

Reviewer #1 (Remarks to the Author):

The authors have addressed all the points I raised in my original review, providing extra analysis and clarifying issues. I thank them for their robust and detailed responses to all points raised.

Reviewer #2 (Remarks to the Author):

With this revised manuscript, the authors have adequately addressed my prior concerns.

Reviewer #3 (Remarks to the Author):

All comments or suggestions made have been addressed by the authors. I have no other comments or suggestions. This article provides interesting information on MTBC, and a useful bioinformatics tool helping users for a better understanding of TB.